# GenEval: A Benchmark Suite for Evaluating Generative Models

## Abstract

Generative models are important for several practical applications, from low level image processing tasks, to model-based planning in robotics. More generally, the study of generative models is motivated by the long-standing endeavor to model uncertainty and to discover structure by leveraging unlabeled data. Unfortunately, the lack of an ultimate task of interest has hindered progress in the field, as there is no established way to compare models and, often times, evaluation is based on mere visual inspection of samples drawn from such models.

In this work, we aim at addressing this problem by introducing a new benchmark evaluation suite, dubbed *GenEval*. GenEval hosts a large array of distributions capturing many important properties of real datasets, yet in a controlled setting, such as lower intrinsic dimensionality, multi-modality, compositionality, independence and causal structure. Any model can be easily plugged for evaluation, provided it can generate samples.

Our extensive evaluation suggests that different models have different strenghts, and that GenEval is a great tool to gain insights about how models and metrics work. We offer GenEval to the community [1] and believe that this benchmark will facilitate comparison and development of new generative models.

## 1 Introduction

Modeling uncertainty is a fundamental problem for machine learning. In unsupervised settings, an ideal model should be able to describe all the possible events consistent with the provided context. In supervised settings, there is often not a single correct output for a given input, and models need to be able to express the space of correct outputs and their relative likelihood.

One standard approach to this problem is to build a probability model of the output space, or more generally, an energy-based model that assigns a score to every possible output. This kind of model is useful for comparing two possibilities, for example. Recently there have been a number of proposed models that do not necessarily assign any score to possible outputs, but using a pseudo-random number generator, produce hallucinations that in some way resemble the true outputs. This kind of model can be useful for examining possibilities.

In this work, we will consider the second kind of model, and take a "generative model" to be any construction that has a method for outputting data points using a pseudo-random number generator. In particular, we do not require "generative models" to be able to compute a probability of a data point. These kinds of models have become popular recently due to several works on Generative Adversarial Networks (GANs) (Goodfellow et al., 2014), that have demonstrated the generation of realistic images (Radford et al., 2015; Karras et al., 2017).

However, this field has struggled to solve its zeroth problem (even as it has made progress on its first). Before the design of "better" models , it is necessary to agree upon good metrics and methodology to evaluate the quality of samples generated by a model. Part of the problem is that the downstream tasks of interest in the case of image generation have not been fully settled. Nevertheless, there have been a few works that make rigorous attempts to quantify the capabilities of these models, restricted to GANs parameterized as convolutional nets in the setting of natural images (Lucic et al., 2017; Huang et al., 2018), and concurrently Kurach et al. (2018).

---

[1] Available at: *coming soon.*

In this work, we expand upon these with a suite of benchmark synthetic distributions that we use to evaluate not only GANs, but also other forms of generative models. We deliberately avoid convolutional networks on images with the aim of decoupling the benefits of various modeling paradigms from domain specific neural architectures. If we believe that generative models should be studied as generic tools for dealing with uncertainty (as opposed to methods of generating images), this decoupling is necessary. Additionally, practical, generic, and effective methods for measuring the distance between distributions in high dimensions with access only to samples do not currently exist. On the other hand, in some cases where one of the distributions has some special structure, we can effectively measure distortion, see an example of this in Appendix sec.A; using simple synthetic distributions allows us to be sure the distributions under consideration have the correct special structures. Analysis of which methods are able to model which kinds of special structures (e.g. independence, causality, multi-modality) is critical for understanding these methods; with "real" data not fully under our control, such analysis can be difficult.

A major contribution of this work is the GenEval toolbox to evaluate models. Thanks to its general API, new models can be easily plugged in and tested using a variety of metrics against popular baseline generative models. Likewise, it's very simple to add new distributions and evaluation metrics to the existing pool to highlight special properties of a model of interest.

## 2 RELATED WORK

Much recent literature on generative modeling of continuous distributions has focused on modeling natural images. Restricting our attention to works aiming at generating samples, as opposed to scoring inputs, many recent models are based on Generative Adversarial Networks (Goodfellow et al., 2014; Arjovsky et al., 2017; Roth et al., 2017; Gulrajani et al., 2017; Denton et al., 2015; Karras et al., 2017), which are capable of generating very realistic high resolution images, a task that had challenged the research community for several decades (Geman and Geman, 1984).

For models trained on natural images, several metrics have been proposed, such as "inception score" (Salimans et al., 2016) and "Frechet inception" distance (Heusel et al., 2017). Unfortunately, these metrics rely on yet another model for evaluation, and are known to only partially correlate with human judgment. To further complicate matters, generation scores not only depend on the metric but also on the choice of architecture of the generator, typically a rather sophisticated convolutional neural network. To summarize, comparison between different generative models is very difficult, and it is often based on visual inspection of a handful of model samples (Denton et al., 2015; Arjovsky et al., 2017).

Several works have attempted to isolate the contribution of the estimation method by working in a controlled setting, e.g., by fitting the model to a mixture of Gaussians in two or three dimensions (Arjovsky et al., 2017; Roth et al., 2017; Gulrajani et al., 2017), or in one dimension (Zaheer et al., 2017). This paper can be seen as a more comprehensive, thorough and extensive study building upon these earlier attempts. While we share similar motivations, we do not restrict our focus to just GANs, but also include other popular models (Kingma and Welling, 2013; Dinh et al., 2016).

Recently, Lucic et al. (2017); Kurach et al. (2018) have systematically compared different variants of GANs trained on natural images. Again, unlike this work, these focus only on GANs. Morover, we restrict our study to purely synthetic distributions with known properties which we leverage at evaluation time (or, from another perspective, we broaden our study to distributions beyond those coming from the image domain).

The choice of metrics for generative models has been intense topic of debate. Most works, overall those based on probabilistic modeling, report performance in terms of log-likelihood of samples generated by the data distribution according to the model. As discussed in earlier work (Theis et al., 2016) and in sec. 5, this is not a good metric in general, as it cannot be estimated by all models, e.g., GANs, and it is very susceptible to the scaling of the input. Interestingly, Lucic et al. (2017) propose a precision/recall metric to evaluate the quality of samples, where precision measures the distance of each model sample to the data manifold and recall measures the distance of each data point to the closest model generation. Unfortunately, such score disregards any weighting of the density, the fact that not all data points are equally likely and similarly, not all model samples have the same likelihood. Recognizing this issue, we borrow the "two-sample test" criterion introduced by Lopez-Paz and

Oquab (2016) and extended by Huang et al. (2018). This two-sample statistic is based on a similar motivation but it operates on actual samples, using implicit Monte Carlo weighting to take into account the density, and obviating the need for an explicit optimization over a data-manifold.

## 3 DISTRIBUTIONS

The benchmark focuses on distributions with support on $\mathbb{R}^d$ for some $d$. Their implementation in GenEval requires two methods: one for computing the log-density at a point, and one for sampling. GenEval implementation of distributions is modular, and allows combining distributions via products, mixtures, causal mappings, and isometric transforms. Next, we briefly describe the distributions we considered in this study.

**Simple distributions**   Many of the distributions will be built from standard families of distributions, such as the *multivariate Gaussian*, and the *Slab* (or uniform) distribution with constant density on a rectanguloid in $\mathbb{R}^d$.

Such distributions can be embedded into a lower dimensional space $\mathbb{R}^q$ with $q < d$, yielding what we call a "flat" Guassian or Slab. For instance, the density function of a flat Slab is defined as:

$$p(x) \propto \chi_R(O^T(x-m)) + \eta(x),$$

where $\chi_R$ is the uniform distribution with support in $R = \{x : 0 \leq x_i \leq r_i, i = 1, \ldots, q\}$, $O$ is a $d \times q$ matrix with orthogonal columns, and $\eta$ is a Gaussian density with small variance.

**Mixtures** Starting with $K$ densities $p_1, \ldots, p_k$ on $\mathbb{R}^d$, and a weighting vecor $w$ on the simplex in $\mathbb{R}^d$, the density for a mixture is: $p(x) = \sum_{i=1}^{K} w_i p_i(x)$. When the bulk of the mass of $p_i$ is disjoint from the bulk of the mass of $p_j$ for all $i \neq j$, the mixture model has a *cluster* structure. We show results with these kinds of distributions in fig. 1, 2 and in Appendix sec. E.

**Distributions with causal structure** Given a function $f : \mathbb{R}^q \mapsto \mathbb{R}^{d-q}, q < d$ and a density $p'$ defined in $\mathbb{R}^q$, we can can get a density with causal structure on $\mathbb{R}^d$ by considering

$$p(x) = p'(x^{\text{ind}})\eta(x^{\text{dep}} - f(x^{\text{ind}})),$$

where $x^{\text{ind}}$ are the independent first $q$ coordinates of $x$ (drawn from a certain distribution), and $x^{\text{dep}}$ is the last $d - q$ dependent coordinates of $x$, and $\eta$ is a noise distribution (for example zero-mean Gaussian with small variance). Note that these kinds of distributions have locally lower dimensional structure than $d$, especially when $d$ is large compared to $q$. We show results with these kinds of distributions in fig. 5.

**Locally low dimensional distributions** We consider three types of locally low dimensional distributions: causal distributions where $f$ above is parameterized by a random fully-connected ReLU net or a random quadratic polynomial (to simulate non-linear manifolds), and an "image-like" distribution of shifted bumps.

For the random neural-networks, we use one-hidden-layer ReLU networks with default PyTorch (Paszke et al., 2017) initializations. For the random quadratic polynomials, we take a sample for a two dimensional distribution and we project it into a $d - 2$ dimensional space with a matrix of normally distributed coefficients, and then concatenate both vectors.

For the shifted bumps distribution, which is similar to the artificial triangle image distribution considered in (Lucic et al., 2017), we take a side-length $s$, a set of radii $r_0, \ldots, r_l$, and consider all shifts of an $r_i \times r_i$ square on a $s \times s$ background. These form a set of $s^2 * l$ points in $\mathbb{R}^{s^2}$; we take a random orthogonal projection of this set to $\mathbb{R}^d$ and place a ball of mass around each projected point. Experiments with these distributions are shown in fig. 3.

**Distributions with independence structures** We can increase the intrinsic dimension of any of these previous distributions by combining them as a product: $p(x) = \Pi_{j=1}^{K} p_i(x)$, where $p_1, \ldots, p_K$ are the basic components which have non-zero support on a disojint subset of coordinates. Note that models are not provided with such information at training time. We report results with product distributions in fig. 4 and in Appendix.

## 4 GENERATIVE MODELS

In this section we first describe some classical baselines, meant to provide upper and lower-bound performances on each distribution, and then some neural-network based models.

**Baselines:** The **oracle** baseline is the data distribution itself. Given a metric assessing the discrepancy between samples drawn from the data and the model distributions (see sec. 5), oracle samples provide a *lower bound* on the error of an ideal model perfectly fitting the data. Ideal metrics would report an error close to 0 for oracle samples.

The next baseline we consider is the **kernel density estimator** (KDE) (Rosenblatt, 1956; Parzen, 1962) which places a Gaussian bump of a certain width around each training point. KDE essentially memorizes the training set, and therefore, it is expected to provide great fitting to the training data but poor generalization, overall in high dimensional spaces where the curse of dimensionality would require a vast amount of training samples to finely cover the data distribution.

Next, we have the multivariate **Gaussian** distribution fitted by maximum likelihood (Pedregosa et al., 2011), meant to provide some sort of *upper bound* on the error, and the **mixture of Gaussians** with $k$ components (MoG$_k$), which we fit by using the expectation-maximization algorithm (Dempster et al., 1977). We fix the number of Gaussians in the mixture to 10 in all experiments.

**Neural Models:** Among the neural models, we consider the **Variational Auto-Encoder** (VAE) (Kingma and Welling, 2013), the **real non-volume preserving** density estimator (RNVP) (Dinh et al., 2016) and **Generative Adversarial Network** (GAN) (Goodfellow et al., 2014) with some of its variants, namely Wasserstein GAN (WGAN) (Arjovsky et al., 2017), WGAN with Gradient Penalty (WGAN-GP) (Gulrajani et al., 2017) and GAN with Noise Regularization (GAN-NR) (Roth et al., 2017). See Appendixfor a brief review of these methods.

## 5 DISTORTION STATISTICS

Since all models must be capable of drawing samples, we measure fitting error in terms of the distortion between two sets of points: samples drawn from the data distribution and from the model distribution. We will discuss several measures of fidelity to a distribution; and *none* of them are generically useful. Measuring the distortion between real-valued distributions in high (or even not-so-high) dimensions continues to be an unsolved problem, especially if we add constraints on computational efficiency. This is one of the reasons to use artificial data with special properties we control: although none of the metrics are generally useful, they can all be useful when the distributions have appropriate structure (see example in Appendix sec. A).

**Optimal Transport** The *Optimal Transport distance* (OT) $u^\rho$ between sets $S$ and $T$ of points in a metric space with metric $\rho$, is defined by:

$$u^\rho(S,T) = \min \sum_{i,j} \lambda_{ij}\rho(s_i,t_j) \;\; \text{s.t.} \; \sum_i \lambda_{ij} = 1 \text{ and } \sum_j \lambda_{ij} = 1 \text{ and } \lambda_{ij} \geq 0 \qquad (1)$$

We use the POT package [2] for computing the optimal transport, and assume $\rho$ is Euclidean $l_2$ for the rest of this work, and drop it from the notation.

Sampling a fixed number of points from two distributions $P$ and $Q$ and computing the OT distance between the samples does not give a distance on distributions (even if it were well defined, it would not even give "distance" 0 from a distribution to itself), but it does correspond to an estimator of the OT distance between $P$ and $Q$. However, in high dimensions, with few samples, this estimator can have possibly counter-intuitive behaviors. For example (Arora et al., 2017), suppose we sample $N$ points $S$ and $N$ points $S'$ from the uniform distribution over the unit sphere in $\mathbb{R}^d$, where $d$ is large enough to see concentration-of-measure effects, and then $N$ points $T$ from $\delta_0$ (the point mass at the origin). If $N$ is small compared to $d$, then the points in $S$ and $S'$ are almost orthogonal, and so have distance roughly $\sqrt{2}$ between them. On the other hand, the distance of any point in $S$ to a point

---

[2]http://pot.readthedocs.io/en/stable/

sampled from $T$ is 1. It is thus likely that $u(S, S') > u(S, T)$ even though $S$ is sampled from the same distribution as $S'$ and $T$ is not.

On the other hand, OT has some pleasing properties. It is continuous with respect to perturbations of the points in the sets. Furthermore, if two sets of points covers the same low-dimensional manifold in a high dimensional space, the OT distance between them will be relatively small.

In the tables below, when using OT to measure the success of a generative model, we will always report the "oracle" OT score given by comparing two sets of samples from the true distribution. When the oracle scores are relatively high, the relative results should be taken cautiously.

**Nearest Neighbor Two-Sample Statistic** Two set of points, $S \sim p_d$ and $T \sim p_g$, are close if the chance that a point from $S$ has nearest neighbor belonging to $T$ is 50% and vice versa. This gives the intuition behind the nearest neighbor two-sample statistic (2S) (Huang et al., 2018). In this work, we use a version similar to the one in Huang et al. (2018). Assume $|S| = |T| = N$. For each point $x$ in $S$ or $T$, define $n(x)$ to be[3]

$$n(x) = \begin{cases} 1, & \text{if } \min_{y \in S}||x - y|| < \min_{y \in T}||x - y|| \\ 0, & \text{otherwise.} \end{cases} .$$

Then, we define the distance as the sum of the deviations from the optimal rate:

$$v(S, T) = \left| 1/2 - \sum_{x \in S} n(x)/N \right| + \left| 1/2 - \sum_{x \in T} (1 - n(x))/N \right| \tag{2}$$

As $N$ gets larger, it becomes easier to distinguish the distributions via this statistic; but different values of $N$ may lead to different quality orderings. As the dimension of the distribution gets higher (for fixed $N$), it becomes harder to distinguish distributions via this statistic. As with OT, we will always show the oracle value for a distribution when showing results of generative models.

**Log-Likelihood** In this work log-likelihood (LL) is estimated on the samples drawn from the model using the (known) data distribution - since we do not require models to necessarily be able to estimate data log-likelihood. This gives a notion of how likely points generated by the model are, regardless of the overall fit. In other words, samples drawn from a model assigning all its mass to the mode of the data distribution will have even higher likelihood that samples drawn from the actual data distribution.

## 5.1 SPECIAL STATISTICS

Our distributions have special structure that we can take advantage of, to better measure success in modeling that distribution.

**Mode coverage** (MC) Assuming a mixture distribution for the data and a uniform distribution over modes, we measure whether samples generated from the model have even coverage of the clusters. We report "mode coverage" (MC) as the perpelxity of the mode assignments: $MC = 2^{H(a)}$, where $H$ is the entropy of the cluster assignment distribution $a$.

**Causal Discovery** Assuming a causal distribution (see sec. 3) and the ability of the model to perform conditional inference of a set of variables given the complementary set, we measure how close (in $l_2$) the recovered dependent coordinates are from what the ground truth values given a set of independent coordinates drawn from the true underlying distribution.

For all the neural models with a latent space, we estimate the missing variables by optimizing over the latent variable with an $l_2$ reconstruction loss over the observed coordinates, starting from a random point. For KDE, we find the nearest point from the training set in the constrained coordinates, and return the remaining coordinates of that training point. For a mixture of Guassians, we take the nearest mean in the constrained coordinates among all components, and then return the maximum-likelihood estimate from that Gaussian of the remaining coordinates. See fig. 5 for results using this metric.

**Independence Test** Measuring the independence of high-dimensional real valued distributions is challenging. In order to get some idea of the ability of the various models to detect independence, we will restrict our attention to distributions that are products of distributions that are easy to vector-quantize, and use categorical tests for independence on the quantized values. That is, suppose we

---

[3]In our implementation, if the quantities $\min_{y \in S}||x - y|| = \min_{y \in T}||x - y||$ we choose $n(x)$ randomly in $\{0, 1\}$.

have a distribution $p = \bigotimes_{i=1}^{K} p_i$ where $p_i$ is supported on the set of coordinates $c_i$. Further suppose we have clustered $p_i$ into clusters $C_{ij}$ for $j \in \{1, ..., n_i\}$, and that we have $L$ samples $X$ from a model. Then, for each coordinate group $c_i$, we make a table of size $n_i \times (n_1 n_2 ... n_{i-1} n_{i+1} ... n_K)$ with the counts $N_{s,t}$ of the number of points in $X$ that landed in each cell. Here, $s$ indexes rows of this table, corresponding to the $n_i$ centroids $C_{i1}, ..., C_{in_i}$ for the points projected down to coordinate group $c_i$, and $t$ indexes columns of the table, corresponding to the product of all other clusters in all other coordinate groups. Denote by $M_t = \sum_s N_{s,t}/L$ and $M^s = \sum_t N_{s,t}/L$; we expect that $N_{s,t}/L \sim M_s M_t$, so we take:

$$\chi^2 = \sum_{s,t} (N_{s,t}/L - M_s M_t)^2 .$$

**Independent OT and 2S** There is a partial remedy for dealing with the weaknesses of the distortion measures in high dimensions for distributions that we know have independence structure, see sec. 3. Suppose that the distribution $p$ we are interested in has support in $\mathbb{R}^d$. Further suppose there are groups of coordinates $I_0, I_1, ...I_L$ partitioning $[1, ..., d]$ such that $p_{I_j} \perp p_{I_i}$ for $i \neq j$. Here $p_I$ is the distribution given by projecting $p$ onto the subspace spanned by the coordinates in $I$. Then, for some distortion measure $u$, we can use the set distance given by:

$$u_{\text{ind}}(S, T) = \sum_{i=1}^{K} u(S_{I_i}, T_{I_i}), \tag{3}$$

where $S_{I_i}$ is the projection of $S$ onto $I_i$ (and likewise for $T$).

In building such a distortion measure, we are using the knowledge of the true distribution. However, because *we* are building the distributions in the benchmark, we are free to use this information in measuring distortion, even if the models should not get access to it during training. Furthermore, note that this kind of distortion measure only can tell the difference between distributions that disagree on the marginals of the groups. Even if it shows no distortion between two distributions, they may still be different. On the other hand, because each group has smaller dimension, the distortion measure is less pre-disposed to show everything being different from everything else.

## 6 GENEVAL

GenEval is written in Python and it consists of three main components: models, distributions, and metrics.

Models must define `fit(X)` and `sample(N)` methods, the former to train the model on a dataset of points $X$, the latter to draw $N$ samples. Models can optionally define a method to conditionally sample data given values for a set of fixed coordinates. Thanks to the general interface and modularity of GenEval, we simply incorporated the original implementations by the authors of the models whenever available. As a result, GenEval does not enforce any specific machine learning framework. For VAE(vae), GAN(gan, b), WGAN(wga, b), WGAN-GP(wga, a) and GAN-NR(gan, a) we used PyTorch (Paszke et al., 2017), while for RNVP(nvp) we used TensorFlow (Abadi et al., 2015). For Mixture of Gaussians and Kernel Density Estimation we employ the scikit-learn (Pedregosa et al., 2011) package.

Each distribution class must define the methods: `sample(N)` and `logprob(X)`. Mixture distributions may also optionally define a method to estimate the most likely cluster assignment to each input data point, and product distributions can specify their components. New distributions can easily be defined via composition directly in configuration files specifying which distribution to test on. For instance, the library contains classes for a slab, a mixture of arbitrary distributions, and affine transform. Given these, one may define on-the-fly a distribution over the surface of a rotated 3D box in some higher dimensional space.

Metrics methods take as input two sets of samples (from the data distribution and from the model) and output a scalar value. Some metrics may optionally take as input a trained model and the data distribution to compute distribution dependent metrics like mode coverage.

Models and distributions are passed as input to GenEval via two Python script configuration files, which specify all their eventual hyper-parameters. Model hyper-parameters can also be specified via

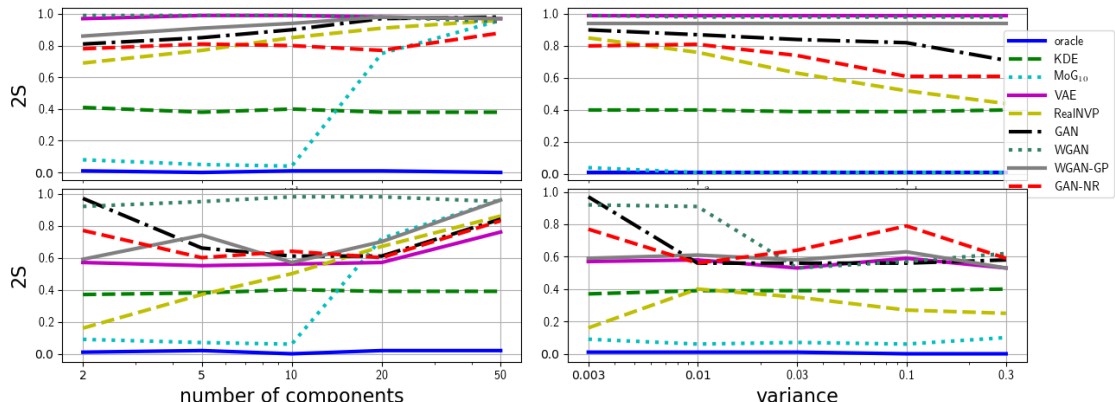

Figure 1: Two Sample test on Mixture Gaussians. Top: each Gaussian component has intrinsic dimensionality equal to 5, with a spherical covariance (in 5d) rotated to a random orientation 50d and non-zero mean placed at random; the ambient dimension is 50. Bottom: each Gaussian component is defined in a 50 dimensional space, and it has spherical covariance and a non-zero mean placed at random in that space. Left: varying the number of components in the mixture from 2 to 50. Right: varying the variance of each component; note that for the flat mixture (Top) only the variance in the intrinsic dimension changes.

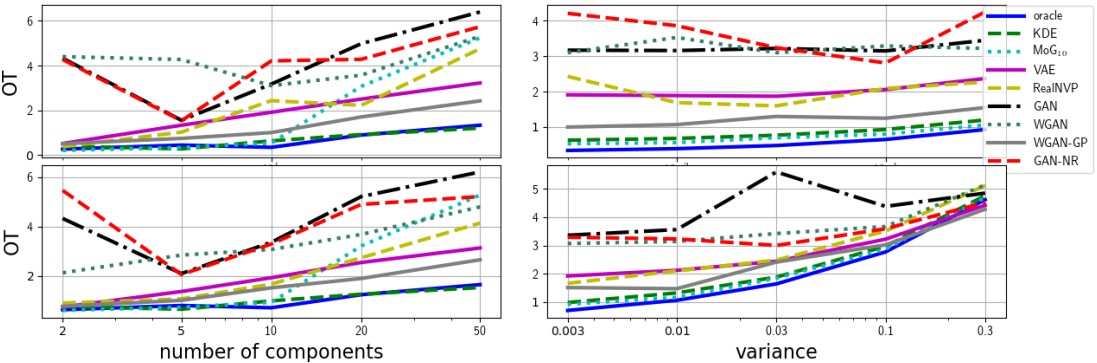

Figure 2: Same as above but using OT for both cross-validation and evaluation.

lists, which are used by GenEval to run a grid search over hyper-parameter values. A user may then run GenEval with configuration files that specify several distributions and several models. GenEval then launches training and evaluation for the cartesian product of all possible combinations of model and distribution and report results on a table, as those shown in the Appendix.

# 7 EXPERIMENTS

We used GenEval to compare the models described in sec. 4 on several distributions. For each distribution, GenEval first runs a grid search over hyper-parameters, see for instance the configuration files in Appendix sec. D. To produce the results in this section for every method we ran a very extensive grid search over hyper-parameters, which reached about $20,000$ configurations for some GAN variants; and was in the thousands for all neural methods. Afterwards, GenEval compares models across all metrics and compiles tables, such as those in Appendix E.

Unless otherwise stated, all experiments we discuss next have used 10,000 training samples, and unless otherwise specified in the figure, 1000 validation and test samples. On the figures showing results with OT, OT was used as a cross-validation metric for the hyperparameter search; and 2S was used for figures showing results with 2S. For all other figures, cross validation was done using 2S.

Fig 1 shows two-sample results when the true distribution is a mixture of Gaussians with varying number of components in the mixture (left), variance (bottom) and intrinsic dimensionality (top versus bottom). First, we observe that model performance is bounded by the oracle, as expected, with a nearly 0 distortion. A mixture of Gaussian with 10 components achieves the lowest distortion when the number of components in the true distribution is less than 10, but distortion degrades

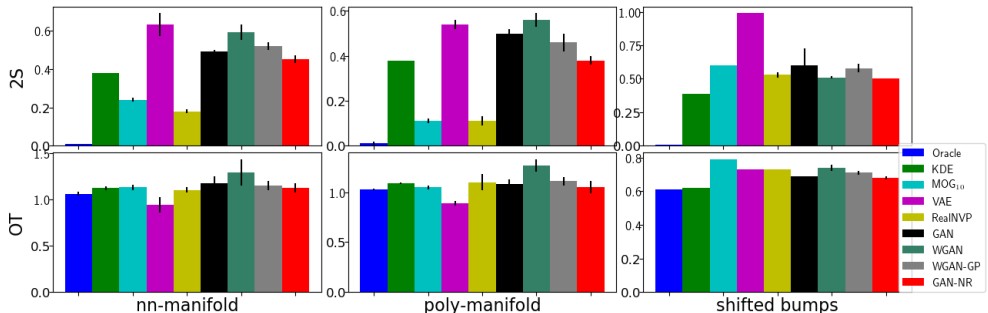

Figure 3: Two-sample statistics and optimal transport on two and three dimensional embedded manifolds in $\mathbb{R}^{50}$.

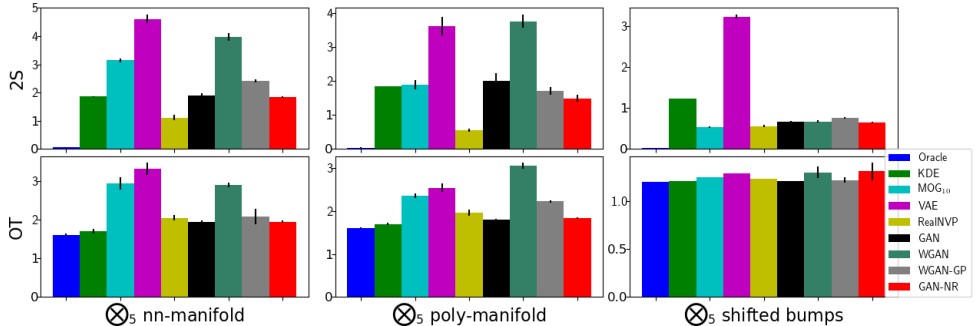

Figure 4: 2S and OT on products of five identical distributions, each of which has support near a two or three dimensional embedded manifold in $\mathbb{R}^{10}$. In this figure, we use $u_{\text{ind}}$ from eq. 3 (using the true groups of independent coordinates) to measure distortion and to validate models.

very rapidly as soon as the true number of components increases. The KDE baseline instead has distortion which is constant with respect to number of components and variance[4]. KDE often leads to competitive distortion compared to neural models, yet much higher than the oracle. Second, in this experiment, GAN variants do not perform reliably better than the original GAN. Third, we observe that none of the neural models perform well, despite the relative simplicity of the distribution. Modeling distributions with more components, tighter variance and lower intrinsic dimensionality makes the learning problem considerably harder for these models. The results are somewhat different if we cross-validate and evaluate according to OT, as shown in fig. 2. With this statistic, there is no performance gap between the oracle and KDE, highlighting the importance of the choice of metric when comparing methods. Oracle performance according to OT degrades as a function of the number of components (left half of Figure 2), and especially as the intrinsic variance increases (bottom right of Figure 2), but not when the intrinsic dimensionality is lower (top right of Figure 2). Thus, as suggested by the example in 5, sampled OT can be an unreliable measure of success when the true distribution fills out space in high dimensions.

Next, we evaluate on various manifold distribution and product of manifold distributions, see fig. 3 and 4. One interesting observation is that OT and 2S do not correlate very well (see discussion in sec. 5). On these tasks, Real NVP performs well in terms of the nearest neighbor two-sample statistic, followed by $\text{MoG}_{10}$, with other methods performing worse. In terms of OT, KDE does best on the products of manifolds, and VAE does the best on the manifolds. In fact, VAE does better than oracle there (see two bottom left plots in fig. 3 and the note in 5), suggesting it is denoising the true distribution.

In the next experiment reported in fig. 5, we analyze the ability of the models to discover causal structure on manifold distributions. We do not report the results of Real NVP here because we cannot

---

[4]This is essentially a function of the number of points used for computing the two-sample statistic vs. the number of training points. The nearest neighbors of real points may be generated or real; KDE is essentially perfect on this side of the distortion statistic. But the neighbors of generated points will almost certainly be generated as the number of test points increases, as the neighbor will have been generated from the Gaussian about the same training point.

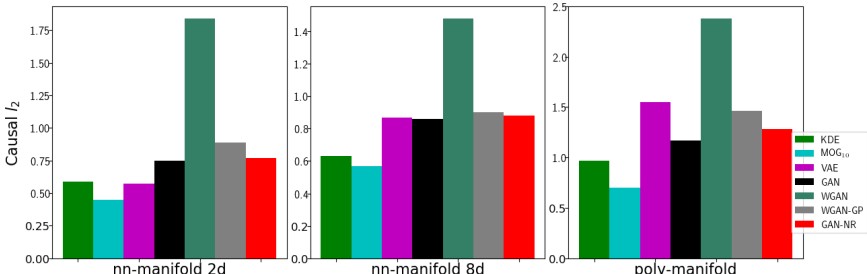

Figure 5: Prediction error of dependent variables in causal distribution, see sec. 3 and 5.1. Hyperparameters are chosen by best OT.

make a conditional estimate in the same way as the other neural models. We observe that all models have hard time beating the $\text{MoG}_{10}$ baseline in this case.

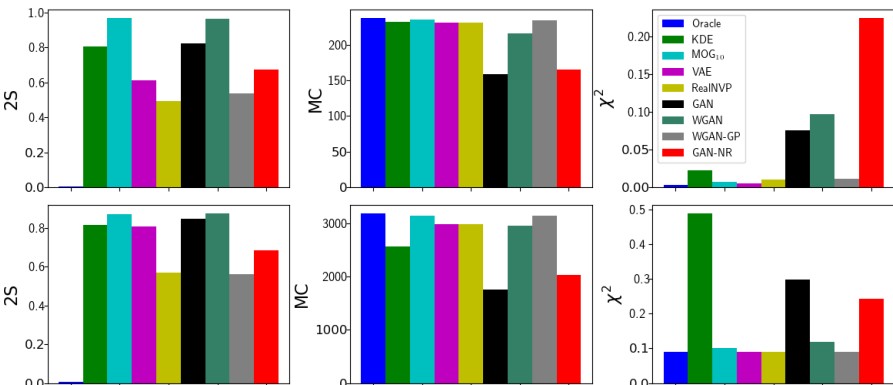

Figure 6: We consider product of 5 (top) and 8 (bottom) distributions, each of which is a mixture of three Gaussians in 12 dimensions. The $\chi^2$ metric, see sec. 5.1, measures the extent by which models have captured the independence structure of the product distribution. Note that higher is better for mode coverage, unlike the other metrics. 2S is computed using 1K samples, while MC and $\chi^2$ use 50K samples

Finally, we look at a product of independent Gaussian mixture distributions in fig. 6. None of the models are able to succesfully fit this distribution, although WGAN-GP and Real NVP do the best. KDE in particular fails to fit this distribution, both in terms of two-sample and in terms of the $\chi^2$ independence test. We can also see that all of the GAN variants improve on the mode coverage of the vanilla GAN.

## 8 DISCUSSION

Perhaps to the delight of neural generative model skeptics, one sees that on almost all distributions and all metrics that we consider, one of KDE or mixture of 10 Gaussians are competitive with (and often superior to) the neural models. These take a tiny fraction of the training cost, have essentially no hyper-parameter sweep, and have simple, well understood fitting routines. However, the neural models, especially GAN, are designed to be used where $l_2$ in the observation space makes little sense as a metric. On most of our examples $l_2$ in the observation space is locally meaningful, and most of our metrics rely on $l_2$ to be (at least) locally meaningful. Thus we consider success at our benchmarks neither necessary nor sufficient for a model to be good at the perhaps more complex tasks for which neural models are designed. We discourage neural model skeptics from dismissing neural models based on these results: worse results on simple tasks against unscaleable baselines better adapted to the simple tasks does not necessarily mean an approach should be abandoned.

On the other hand, neural generative model boosters should take heed of these results. They show that on datasets that are not images, with networks that are not convolutional networks, neural models do not do well. Moreover, specific to GANs, our results corroborate the results of Zaheer et al. (2017);

Arora and Zhang (2017), and suggest that GANs do not necessarily learn to sample from the target distribution. Furthermore, some protocols purported to improve GANs in the image setting do not reliably improve results across the metrics and datasets shown here (although cluster coverage does seem to be reliably improved). This is natural, as practitioners have spent much effort tuning models for performance in the image domain, and of course neural models will do poorly without well adapted architectures [5]. Nevertheless, most works trying to improve or understand GANs (empirically or theoretically) discuss the training protocol independently of the relationships between the inductive biases of the neural architectures and the properties of the distributions to be modeled. Our results here suggest that these relationships cannot be ignored when studying GANs.

We also see that while no single neural model is all-around superior to the others, Real NVP does do better than the other models in many cases, especially in terms of the two-sample statistic. However, it does not dominate, for example doing worse at products of manifolds in terms of OT, see fig. 4, and worse on mixtures of Gaussians with large numbers of components, see fig. 1[6].

We can also see other quirks of the models. For example, we can see that "denoising" effects of a VAE can make it appear to be a good model on some datasets and metrics, but poor in others. It is one of the better performing neural models in fig. 5 when measuring causal error, showing it has learned the manifold and the functional dependence. But at the same time, it is bad in terms of twosample (fig. 3); perhaps because it is collapsing the two free components in the output. More generally, these kinds of effects show that in general, it is important to consider multiple metrics. The gold standard should be a success in a downstream task of interest, but in the absence of such a task, looking at a single metric to judge success can be misleading.

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

## A    BLOCK INDEPENDENCE AND DISTORTION MEASURE RESOLUTION

We give a simple example showing how eq. 3 can increase the accuracy, or more precisely, the resolution of a distortion measure. We take a product of 10 mixtures of Gaussians in $\mathbb{R}^{12}$, each with 3 components, with modes randomly distributed over the unit sphere in $\mathbb{R}^{12}$, as in fig. 6. The overall ambient dimension is 120.

Measuring OT against a single best fit Gaussian, which is clearly a poor model of the original distribution, without leveraging the groups structure (see again eq. 3), gives a distortion of 7.3 for the oracle samples versus 8.6 for the samples from the Gaussian. If we do leverage information about the independence structure among the groups, we get a distortion of 5.9 for the oracle versus 19.3 for the Gaussian. Raising the number of mixtures in the product to 30 gives 17.4 vs 17.6 without groups and 17.5 vs 54.5 with.

It is therefore essential even in relatively low dimensions to leverage the special structure of the distributions in order to assess the real quality of the sameples generated by a model.

## B    OTHER METRICS

In addition to the metrics introduced in sec. 5,we also considered the Hausdorff distance.

**Hausdorff Distance**    The *Hausdorff distance* (H) $h_\infty^\rho$ between sets $S$ and $T$ of points in a metric space with metric $\rho$, is defined by:

$$h_\infty(S,T) = \max_{s \in S} \min_{t \in T} \rho(s,t) + \max_{t \in T} \min_{s \in S} \rho(s,t) \tag{4}$$

As in the case of optimal transport, this defines a metric between sets; but when we sample from the distributions and compare the distance between the samples, we do not get a metric on the distributions. We will denote by $H_\infty(P,Q;N)$ the statistic obtained by sampling $N$ points from $P$, $N$ points from $Q$, and computing the Hasudorff distance between the sampled points. We will show results with an average version of this statistic: by $H_2(P,Q;N)$ we take the average minimal square distance, instead of the max. That is, define:

$$h_2(S,T)^2 = \frac{1}{N}\left(\sum_{s \in S} \min_{t \in T} \rho(s,t)^2 + \sum_{t \in T} \min_{s \in S} \rho(s,t)^2\right), \tag{5}$$

and set the statistic $H_2(P,Q;N)$ to be $h_2(S,T)$ for $N$ points $S$ sampled from $P$ and $N$ points $T$ sampled from $Q$. Note that $h_2$ is not even a metric on sets, as it does not satisfy the triangle inequality.

The Hausdorff statistics should be taken as a measure of the difference of support between distributions. That is, as the number of sampled points gets large, two distribution with the same support (but perhaps very different densities on that support) will have relatively small difference in the Hausdorff statistic.

# C ANALYSES OF TWO-SAMPLE TEST

The Two-Sample Test depends on the number of points sampled from the distributions. We analyze how number of test points and number of training points affects the metric. We use random quadratic polynomial distribution and a distributions of shifted bumps as shown in fig. 7 and as described in sec. 3. For both distributions we observe a similar pattern. As the number of points goes up the errors also goes up for all the models. However, the degradation is much more prominent for KDE. The same applied to the size of the training set: neural models may generalize from less amount of training data.

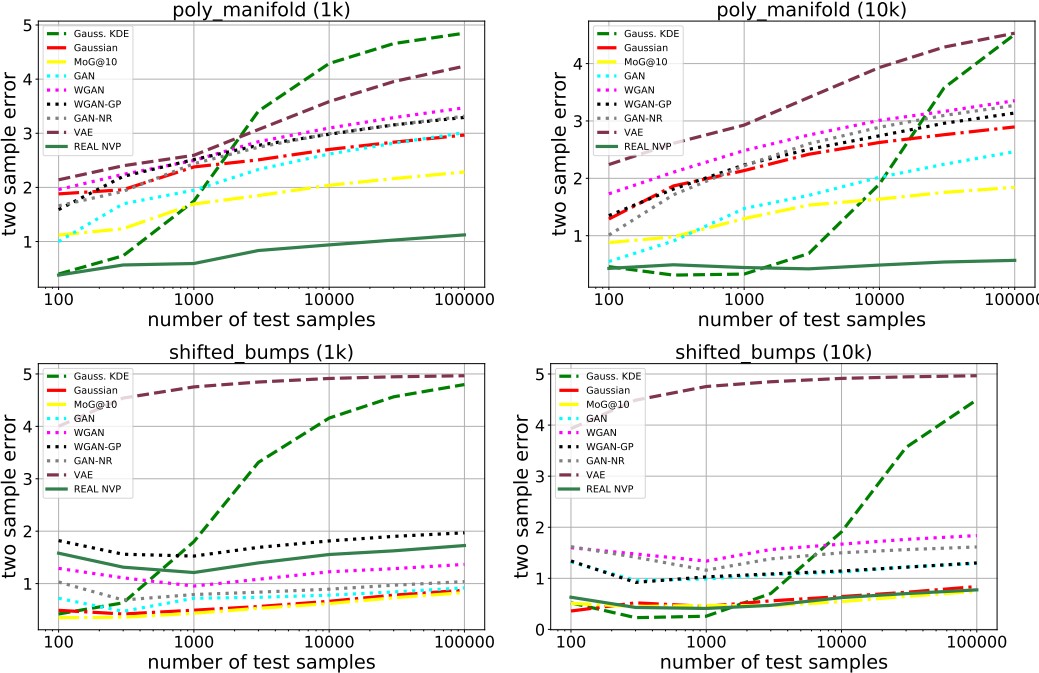

Figure 7: Two-Sample test for different number of test points for a random quadratic polynomial distribution (top) and for a distribution of shifted bumps (bottom), when training with 1k (left) and 10k (right) samples. Lower is better.

## D    HYPER-PARAMETER SEARCH

As mentioned in the section 6, model configurations for each experiment are defined in configuration files. Each file defines a `MODELS` variable with a list of dictionaries describing the model. Each dictionary has three keys: `name`, `model` (refers to a model class), and `args`. A user may want to re-use the same `model` in several cases to evaluate impact of different subsets of parameters. E.g., all flavors of the GAN models implemented in a single class and evaluated separately. Hyper parameters for each case are defined in `args` dictionary that maps parameter to one or more possible values.

All our models use multilayer perceptron as the base architecture. The default activation function is ReLU, but in order to help the model to sample from a mixture of gaussians, we also considered $q$-ary multi maxout activation that is a max pooling over groups of $q$ units. The activation function is determined by `maxout` parameter. Zero stands for ReLU, while positive values stand of $q$-ary multi maxout. Parameter `--nz` defines the size of the latent space. By default GAN models use RMSProp optimizer unless `--adam` is specified. RealNVP and VAE use Adam optimizer by default. We followed the choice in the original implementations here. (Arjovsky et al., 2017) noticed that for better results in training WGAN several discriminator updates should be done per generator updates. Besides, discriminator should get 100 extra updates for first 25 steps and every 500th step. We captures these heuristics in `--Diters` and `--boost_discriminator` flags.

For RealNVP we use only *channel* based masking with 8 coupling layers (defined by `--chain_length`). Each layer is implemented by a multilayer perceptron with optional batch and weight normalization.

Below we show the complete configuration file used for multimodal distributions.

```
import models

MODELS = [
    {
        'name': 'KDE',
        'args': {
            '--kernel': 'gaussian',
            '--bandwidth': [0.001, 0.01, 0.1]
        },
        'model': models.KDE
    },
    {
        'name': 'Gaussian',
        'args': {
            '--n_components': 1,
        },
        'model': models.GaussianMixture
    },
    {
        'name': 'MOG',
        'args': {
            '--n_components': 10,
        },
        'model': models.GaussianMixture
    },
    {
        'name': 'VAE',
        'args': {
            '--nupdates': [1000, 3000, 10000],
            '--num_hidden_layers': [1, 2],
            '--encoder_num_hidden_layers': [1, 2],
            '--hidden_size': [100, 500],
            '--nz': [10, 30, 100],
            '--Emaxout_q': [0, 10],
            '--Dmaxout_q': [0, 10],
            '--lr': [3e-3, 1e-3, 3e-4, 1e-4],
            '--batch_size': 64,
        },
        'model': models.VAE
    },
    {
        'name': 'RealNVP',
        'args': {
            '--mlp_hidden_layers': 1,
            '--mlp_hidden_size': [100, 300],
            '--chain_length': [4, 8],
            '--train_steps': [1000, 3000, 10000],
            '--learning_rate': [1e-2, 3e-3, 1e-3, 3e-4],
            '--batch_size': 64,
            '--weight_norm': [0, 1],
            '--use_batch_norm': [0, 1],
            '--maxout_q': [0, 10],
            '--optimizer': 'adam',
        },
        'model': models.RealNvp,
    },
    {
        'name': 'GAN',
        'args': {
            '--loss': 'gan',
            '--nupdates': [1000, 3000, 10000],
            '--num_hidden_layers': [1, 2],
            '--d_num_hidden_layers': [1, 2],
            '--d_hidden_size': [50, 500],
            '--Gmaxout': [0, 10],
            '--Dmaxout': [0, 10],
            '--nz': [10, 100],
            '--cuda': True,
            '--hidden_size': [100, 500],
            '--batchSize': 64,
            '--lr': [1e-3, 3e-4, 1e-4],
            '--Diters': [1, 2, 5],
            '--adam': True,
        },
        'model': models.GAN
    },
    {
        'name': 'WGAN',
        'args': {
            '--loss': 'wgan',
```

```
                    '—boost_discriminator ': True ,
                    '—nupdates ': [1000, 3000, 10000],
                    '—num_hidden_layers ': [1, 2],
                    '—d_num_hidden_layers ': [1, 2],
                    '—d_hidden_size ': [50, 500],
                    '—nz ': [10, 100],
                    '—Gmaxout ': [0, 10],
                    '—Dmaxout ': [0, 10],
                    '—hidden_size ': [100, 500],
                    '—cuda ': True ,
                    '—batchSize ': 64,
                    '—lr ': [1e−3, 3e−4, 1e−4],
                    '—Diters ': [1, 2, 5],
                    '—adam ': True ,
                },
            'model ': models .GAN
        },
        {
            'name ': 'WGAN—GP ' ,
            'args ': {
                    '—loss ': 'wgan—gp ',
                    '—num_hidden_layers ': [1, 2],
                    '—d_num_hidden_layers ': [1, 2],
                    '—d_hidden_size ': [50, 500],
                    '—nupdates ': [1000, 3000, 10000],
                    '—Gmaxout ': [0, 10],
                    '—Dmaxout ': [0, 10],
                    '—nz ': [10, 100],
                    '—hidden_size ': [100, 500],
                    '—cuda ': True ,
                    '—batchSize ': 64,
                    '—lr ': [1e−3, 3e−4, 1e−4],
                    '—Diters ': [1, 2, 5],
                    '—adam ': True ,
                },
            'model ': models .GAN
        },
        {
            'name ': 'GAN—NR ' ,
            'args ': {
                    '—loss ': 'gan ',
                    '—nupdates ': [1000, 3000, 10000],
                    '—num_hidden_layers ': [1, 2],
                    '—d_num_hidden_layers ': [1, 2],
                    '—d_hidden_size ': [50, 500],
                    '—Gmaxout ': [0, 10],
                    '—Dmaxout ': [0, 10],
                    '—nz ': [10, 100],
                    '—hidden_size ': [100, 500],
                    '—cuda ': True ,
                    '—batchSize ': 64,
                    '—lr ': [1e−3, 3e−4, 1e−4],
                    '—Diters ': [1, 2, 5],
                    '—adam ': True ,
                    '—disc_reg_weight ': [1.0, 0.1, 0.01, 0.001],
                },
            'model ': models .GAN
        },
]
```

## E  MULTIMODAL DISTRIBUTIONS

In this section, we report detailed results using various mixture of Gaussians. "ad" means ambient dimension, "id" means intrinsic dimensions where the MoGs are embedded, "mt" specifies the kind of mean allocation (on the circle "C" or at random "R"), "r" refers to the variance and "MoG" to the number of components in the mixture.

| | oracle | KDE | Gaussian | MOG | VAE | RealNVP | GAN | WGAN | WGAN-GP | GAN-NR |
|---|---|---|---|---|---|---|---|---|---|---|
| MoG2_ad50_id5_mtC_r0.03 | 0.089 | 0.114 | 0.536 | 0.059 | 0.316 | 0.196 | 0.139 | 0.316 | 0.154 | 0.111 |
| MoG5_ad50_id5_mtC_r0.03 | 0.104 | 0.089 | 0.531 | 0.108 | 0.770 | 0.197 | 0.407 | 0.484 | 0.154 | 0.142 |
| MoG10_ad50_id5_mtC_r0.03 | 0.106 | 0.115 | 0.453 | 0.112 | 0.654 | 0.233 | 0.160 | 0.314 | 0.144 | 0.158 |
| MoG20_ad50_id5_mtC_r0.03 | 0.127 | 0.144 | 0.426 | 0.151 | 0.674 | 0.276 | 0.170 | 0.918 | 0.165 | 0.162 |
| MoG50_ad50_id5_mtC_r0.03 | 0.152 | 0.150 | 0.421 | 0.160 | 0.648 | 0.298 | 0.151 | 0.456 | 0.180 | 0.222 |
| MoG10_ad50_id5_mtC_r0.1 | 0.160 | 0.168 | 0.499 | 0.176 | 0.678 | 0.323 | 0.243 | 0.469 | 0.186 | 0.275 |
| MoG10_ad50_id5_mtC_r0.3 | 0.250 | 0.258 | 0.611 | 0.280 | 0.766 | 0.404 | 0.391 | 0.581 | 0.289 | 0.306 |
| MoG10_ad50_id5_mtC_r1.0 | 0.429 | 0.440 | 0.885 | 0.436 | 0.962 | 0.553 | 0.487 | 0.815 | 0.491 | 0.492 |
| MoG10_ad50_id5_mtC_r3.0 | 0.721 | 0.741 | 1.369 | 0.802 | 1.326 | 0.852 | 0.837 | 1.136 | 0.906 | 0.782 |
| MoG2_ad50_id5_mtR_r0.03 | 0.257 | 0.392 | 2.774 | 0.103 | 0.477 | 0.239 | 5.556 | 8.248 | 0.336 | 5.424 |
| MoG5_ad50_id5_mtR_r0.03 | 0.435 | 0.275 | 5.172 | 0.252 | 1.204 | 1.084 | 1.346 | 4.635 | 0.638 | 1.340 |
| MoG10_ad50_id5_mtR_r0.03 | 0.338 | 0.632 | 6.353 | 0.313 | 1.816 | 1.651 | 2.887 | 2.736 | 1.000 | 4.268 |
| MoG20_ad50_id5_mtR_r0.03 | 0.886 | 0.911 | 7.047 | 2.919 | 2.607 | 2.128 | 4.869 | 3.239 | 1.890 | 3.787 |
| MoG50_ad50_id5_mtR_r0.03 | 1.328 | 1.202 | 7.523 | 5.254 | 3.199 | 4.149 | 6.429 | 5.750 | 2.256 | 5.941 |
| MoG10_ad50_id5_mtR_r0.1 | 0.391 | 0.682 | 6.336 | 0.367 | 1.935 | 1.572 | 2.862 | 4.341 | 0.887 | 3.928 |
| MoG10_ad50_id5_mtR_r0.3 | 0.478 | 0.766 | 6.283 | 0.622 | 1.662 | 1.640 | 2.891 | 3.175 | 1.368 | 2.764 |
| MoG10_ad50_id5_mtR_r1.0 | 0.648 | 0.930 | 6.264 | 0.835 | 2.095 | 2.194 | 2.527 | 3.534 | 1.148 | 2.826 |
| MoG10_ad50_id5_mtR_r3.0 | 0.926 | 1.201 | 6.291 | 1.094 | 2.404 | 2.421 | 3.401 | 2.989 | 1.796 | 3.536 |
| MoG2_ad50_id50_mtC_r0.03 | 0.452 | 0.473 | 0.689 | 0.426 | 0.488 | 0.521 | 1.009 | 1.111 | 1.054 | 0.495 |
| MoG5_ad50_id50_mtC_r0.03 | 0.464 | 0.455 | 0.668 | 0.477 | 0.772 | 0.517 | 0.883 | 0.639 | 0.428 | 0.443 |
| MoG10_ad50_id50_mtC_r0.03 | 0.464 | 0.467 | 0.624 | 0.463 | 0.797 | 0.536 | 0.827 | 0.678 | 0.613 | 0.425 |
| MoG20_ad50_id50_mtC_r0.03 | 0.469 | 0.476 | 0.613 | 0.465 | 0.798 | 0.520 | 0.450 | 0.573 | 0.462 | 0.454 |
| MoG50_ad50_id50_mtC_r0.03 | 0.471 | 0.474 | 0.613 | 0.465 | 0.825 | 0.503 | 0.442 | 0.486 | 0.404 | 0.436 |
| MoG10_ad50_id50_mtC_r0.1 | 0.824 | 0.826 | 0.919 | 0.825 | 0.989 | 0.853 | 0.750 | 0.847 | 0.771 | 0.952 |
| MoG10_ad50_id50_mtC_r0.3 | 1.406 | 1.407 | 1.451 | 1.404 | 1.487 | 1.417 | 1.383 | 1.282 | 1.253 | 1.289 |
| MoG10_ad50_id50_mtC_r1.0 | 2.512 | 2.511 | 2.520 | 2.508 | 2.223 | 2.477 | 2.267 | 2.148 | 2.179 | 2.271 |
| MoG10_ad50_id50_mtC_r3.0 | 4.268 | 4.263 | 4.256 | 4.258 | 3.730 | 4.188 | 3.646 | 3.644 | 3.653 | |
| MoG2_ad50_id50_mtR_r0.03 | 0.619 | 0.749 | 2.808 | 0.600 | 0.758 | 1.033 | 6.274 | 2.061 | 0.805 | 5.359 |
| MoG5_ad50_id50_mtR_r0.03 | 0.791 | 0.640 | 5.142 | 0.595 | 1.550 | 1.090 | 1.949 | 2.242 | 1.024 | 1.866 |
| MoG10_ad50_id50_mtR_r0.03 | 0.703 | 0.980 | 6.302 | 0.884 | 1.794 | 1.475 | 2.977 | 3.293 | 1.352 | 3.296 |
| MoG20_ad50_id50_mtR_r0.03 | 1.227 | 1.251 | 6.981 | 3.111 | 2.558 | 2.514 | 5.092 | 3.657 | 1.742 | 4.159 |
| MoG50_ad50_id50_mtR_r0.03 | 1.642 | 1.524 | 7.488 | 5.298 | 3.184 | 3.805 | 6.255 | 4.636 | 2.783 | 5.328 |
| MoG10_ad50_id50_mtR_r0.1 | 1.059 | 1.321 | 6.241 | 1.096 | 2.250 | 1.944 | 2.808 | 3.565 | 1.552 | 2.233 |
| MoG10_ad50_id50_mtR_r0.3 | 1.637 | 1.876 | 6.256 | 1.816 | 2.573 | 2.427 | 5.595 | 3.043 | 2.357 | 2.448 |
| MoG10_ad50_id50_mtR_r1.0 | 2.771 | 2.969 | 6.481 | 2.861 | 3.145 | 3.422 | 3.989 | 3.626 | 2.898 | 3.718 |
| MoG10_ad50_id50_mtR_r3.0 | 4.617 | 4.764 | 7.254 | 4.664 | 4.457 | 5.114 | 4.710 | 4.790 | 4.331 | 4.413 |

Table 1: Mixture distributions with 10000 training points using OT metric.

| | oracle | KDE | Gaussian | MOG | VAE | RealNVP | GAN | WGAN | WGAN-GP | GAN |
|---|---|---|---|---|---|---|---|---|---|---|
| MoG2_ad50_id5_mtC_r0.03 | 253.824 | 231.287 | -170250.471 | 231.369 | -65887.252 | -6900.901 | -57.331 | -53773.787 | -6491.264 | -4697 |
| MoG5_ad50_id5_mtC_r0.03 | 252.981 | 230.307 | -153160.807 | 230.445 | -253697.262 | -15290.232 | -3945.600 | -112854.365 | -6420.108 | -3676 |
| MoG10_ad50_id5_mtC_r0.03 | 252.218 | 229.704 | -115759.422 | 229.747 | -191527.681 | -23915.894 | -3849.664 | -52946.030 | -5687.075 | -6432 |
| MoG20_ad50_id5_mtC_r0.03 | 251.452 | 228.862 | -104456.734 | -6866.577 | -197133.338 | -32354.515 | -6436.782 | -341928.288 | -8642.488 | -6278 |
| MoG50_ad50_id5_mtC_r0.03 | 250.456 | 228.139 | -102708.032 | -5943.212 | -186868.389 | -32109.117 | -7990.157 | -71896.103 | -4529.060 | -7151 |
| MoG10_ad50_id5_mtC_r0.1 | 249.208 | 226.695 | -129044.001 | 226.752 | -191487.241 | -60656.093 | -8093.914 | -78326.775 | -7003.240 | -12623 |
| MoG10_ad50_id5_mtC_r0.3 | 246.462 | 223.948 | -166966.672 | -9825.265 | -209477.349 | -36074.078 | -34197.371 | -126956.270 | -15022.751 | -7276 |
| MoG10_ad50_id5_mtC_r1.0 | 243.452 | 220.938 | -297879.223 | 220.981 | -248019.344 | -57297.815 | -17369.215 | -200859.027 | -27847.380 | -8660 |
| MoG10_ad50_id5_mtC_r3.0 | 240.705 | 218.192 | -663150.864 | -60010.239 | -217340.400 | -58365.110 | -36044.445 | -321284.461 | -30625.161 | -47172 |
| MoG2_ad50_id5_mtR_r0.03 | 253.824 | 231.287 | -4706385.463 | 231.316 | -426133.306 | -62711.711 | -102122.087 | -16675610.717 | -181595.947 | -86721 |
| MoG5_ad50_id5_mtR_r0.03 | 252.981 | 230.307 | -13034230.862 | 230.441 | -1435202.737 | -390003.663 | -954518.627 | -11187066.420 | -173564.214 | -924853 |
| MoG10_ad50_id5_mtR_r0.03 | 252.218 | 229.704 | -18513951.173 | 229.755 | -2318297.079 | -1407379.654 | -3477965.805 | -4363909.568 | -537268.899 | -394392 |
| MoG20_ad50_id5_mtR_r0.03 | 251.452 | 228.862 | -22307511.884 | -5061117.625 | -3678013.955 | -1984178.067 | -581505.899 | -4830178.304 | -1608774.782 | -2810725 |
| MoG50_ad50_id5_mtR_r0.03 | 250.456 | 228.139 | -24634424.153 | -12267927.710 | -5016707.836 | -8313216.364 | -12919610.973 | -15440539.432 | -1798768.727 | -853523 |
| MoG10_ad50_id5_mtR_r0.1 | 249.208 | 226.695 | -18525897.945 | 226.741 | -2157861.216 | -1320055.362 | -3339959.070 | -9290551.421 | -793372.647 | -528493 |
| MoG10_ad50_id5_mtR_r0.3 | 246.462 | 223.948 | -18699702.888 | 223.986 | -2082989.824 | -1435225.157 | -3352350.916 | -5709800.866 | -748740.215 | -142109 |
| MoG10_ad50_id5_mtR_r1.0 | 243.452 | 220.938 | -18828035.613 | 221.002 | -2682950.001 | -1463626.337 | -216242.356 | -6440277.438 | -512050.242 | -1850674 |
| MoG10_ad50_id5_mtR_r3.0 | 240.705 | 218.192 | -19192966.205 | 218.240 | -2769326.615 | -1391332.768 | -3652568.951 | -4575101.256 | -690167.089 | -244264 |
| MoG2_ad50_id50_mtC_r0.03 | 73.628 | 73.576 | 8.073 | 73.656 | 72.604 | 70.636 | 84.551 | -18.458 | -65.469 | 84 |
| MoG5_ad50_id50_mtC_r0.03 | 72.724 | 72.700 | 17.434 | 72.743 | 15.856 | 62.414 | -38.952 | 31.335 | 85.773 | 81 |
| MoG10_ad50_id50_mtC_r0.03 | 72.095 | 72.011 | 31.006 | 72.045 | 11.079 | 56.194 | 1.525 | 24.857 | 60.130 | 89 |
| MoG20_ad50_id50_mtC_r0.03 | 71.276 | 71.234 | 33.946 | 70.386 | 8.195 | 60.910 | 81.715 | 45.758 | 78.979 | 80 |
| MoG50_ad50_id50_mtC_r0.03 | 70.864 | 70.936 | 34.300 | 70.844 | 2.004 | 62.431 | 85.098 | 69.406 | 88.844 | 83 |
| MoG10_ad50_id50_mtC_r0.1 | 42.003 | 41.926 | 30.298 | 41.679 | 38.911 | 38.265 | 56.746 | 45.851 | 50.840 | 48 |
| MoG10_ad50_id50_mtC_r0.3 | 14.726 | 14.646 | 11.334 | 14.641 | 27.678 | 13.503 | 19.792 | 29.601 | 27.557 | 26 |
| MoG10_ad50_id50_mtC_r1.0 | -14.799 | -14.891 | -15.546 | -14.922 | 7.104 | -13.602 | 5.036 | 4.401 | 1.838 | 5 |
| MoG10_ad50_id50_mtC_r3.0 | -41.662 | -41.765 | -41.811 | -41.771 | -17.932 | -40.008 | -21.709 | -23.253 | -22.324 | -22 |
| MoG2_ad50_id50_mtR_r0.03 | 73.628 | 73.576 | -1725.184 | 73.654 | -107.994 | 38.340 | -217.560 | -466.513 | 11.341 | -14 |
| MoG5_ad50_id50_mtR_r0.03 | 72.724 | 72.700 | -4787.562 | 72.734 | -423.206 | -35.199 | -298.758 | -983.164 | -103.058 | -287 |
| MoG10_ad50_id50_mtR_r0.03 | 72.095 | 72.011 | -6885.307 | 72.051 | -862.853 | -208.406 | 22.882 | -2321.356 | -389.596 | -465 |
| MoG20_ad50_id50_mtR_r0.03 | 71.263 | 71.227 | -8293.821 | -1950.515 | -1162.677 | -904.543 | -99.933 | -2659.067 | -306.468 | -472 |
| MoG50_ad50_id50_mtR_r0.03 | 70.270 | 70.336 | -9223.065 | -4879.046 | -1742.290 | -2052.524 | -5468.518 | -3669.330 | -702.292 | -534 |
| MoG10_ad50_id50_mtR_r0.1 | 41.996 | 41.919 | -2059.011 | 41.960 | -253.427 | -71.988 | 29.173 | -711.750 | -34.268 | 16 |
| MoG10_ad50_id50_mtR_r0.3 | 14.531 | 14.456 | -684.216 | 14.491 | -68.693 | -23.838 | -28.428 | -140.817 | -30.060 | 16 |
| MoG10_ad50_id50_mtR_r1.0 | -15.569 | -15.642 | -222.121 | -15.611 | -23.774 | -34.471 | -42.190 | -45.174 | -13.998 | -21 |
| MoG10_ad50_id50_mtR_r3.0 | -43.034 | -43.118 | -110.474 | -43.072 | -31.757 | -50.787 | -34.440 | -41.842 | -31.362 | -32 |

Table 2: Mixture distributions with 10000 training points using log-likelihood metric.

| | oracle | KDE | Gaussian | MOG | VAE | RealNVP | GAN | WGAN | WGAN-GP | GAN-NR |
|---|---|---|---|---|---|---|---|---|---|---|
| MoG2_ad50_id5_mtC_r0.03 | 0.237 | 0.258 | 57.920 | 0.233 | 26.330 | 6.417 | 0.627 | 22.517 | 7.451 | 5.940 |
| MoG5_ad50_id5_mtC_r0.03 | 0.287 | 0.318 | 44.034 | 0.294 | 56.186 | 8.342 | 4.037 | 31.937 | 5.233 | 3.522 |
| MoG10_ad50_id5_mtC_r0.03 | 0.361 | 0.391 | 36.817 | 0.368 | 44.635 | 9.226 | 2.561 | 16.402 | 3.961 | 3.021 |
| MoG20_ad50_id5_mtC_r0.03 | 0.446 | 0.484 | 35.249 | 2.085 | 46.903 | 13.206 | 3.322 | 87.685 | 5.983 | 2.621 |
| MoG50_ad50_id5_mtC_r0.03 | 0.612 | 0.653 | 35.234 | 2.084 | 44.958 | 13.368 | 6.592 | 21.481 | 3.420 | 3.515 |
| MoG10_ad50_id5_mtC_r0.1 | 1.112 | 1.208 | 38.380 | 1.108 | 43.889 | 245.109 | 6.054 | 22.469 | 4.014 | 6.244 |
| MoG10_ad50_id5_mtC_r0.3 | 3.256 | 3.541 | 46.841 | 7.086 | 50.803 | 13.608 | 14.102 | 36.612 | 7.781 | 5.641 |
| MoG10_ad50_id5_mtC_r1.0 | 10.756 | 11.707 | 85.637 | 10.767 | 80.277 | 27.669 | 16.099 | 69.350 | 17.915 | 14.710 |
| MoG10_ad50_id5_mtC_r3.0 | 32.184 | 35.043 | 205.049 | 53.285 | 179.895 | 51.730 | 56.791 | 140.923 | 47.643 | 42.885 |
| MoG2_ad50_id5_mtR_r0.03 | 0.237 | 0.258 | 1688.426 | 0.236 | 382.969 | 211.919 | 7457.907 | 4194.667 | 246.576 | 7166.328 |
| MoG5_ad50_id5_mtR_r0.03 | 0.287 | 0.318 | 3340.899 | 0.295 | 712.163 | 333.198 | 418.922 | 4288.632 | 158.407 | 404.435 |
| MoG10_ad50_id5_mtR_r0.03 | 0.361 | 0.391 | 4480.731 | 0.368 | 951.298 | 705.150 | 1976.482 | 1415.009 | 373.751 | 4292.463 |
| MoG20_ad50_id5_mtR_r0.03 | 0.446 | 0.484 | 5498.846 | 1734.484 | 1204.290 | 884.249 | 4191.913 | 1494.503 | 688.596 | 2085.471 |
| MoG50_ad50_id5_mtR_r0.03 | 0.612 | 0.653 | 6568.851 | 3224.388 | 1464.644 | 3181.403 | 4096.990 | 5522.390 | 764.674 | 3614.357 |
| MoG10_ad50_id5_mtR_r0.1 | 1.112 | 1.208 | 4436.725 | 1.118 | 852.832 | 725.129 | 1964.068 | 3033.507 | 482.750 | 2128.474 |
| MoG10_ad50_id5_mtR_r0.3 | 3.256 | 3.541 | 4389.701 | 3.249 | 851.563 | 722.119 | 1948.733 | 1964.690 | 424.794 | 3194.469 |
| MoG10_ad50_id5_mtR_r1.0 | 10.756 | 11.707 | 4326.580 | 10.872 | 981.120 | 710.919 | 2978.250 | 1837.948 | 347.022 | 709.904 |
| MoG10_ad50_id5_mtR_r3.0 | 32.184 | 35.043 | 4341.083 | 32.344 | 1002.917 | 716.591 | 2654.381 | 1298.073 | 389.408 | 2183.150 |
| MoG2_ad50_id50_mtC_r0.03 | 20.799 | 21.094 | 62.118 | 20.821 | 27.443 | 22.756 | 18.180 | 88.884 | 125.598 | 17.226 |
| MoG5_ad50_id50_mtC_r0.03 | 21.699 | 22.049 | 51.226 | 21.732 | 46.620 | 28.345 | 148.114 | 48.010 | 20.331 | 19.458 |
| MoG10_ad50_id50_mtC_r0.03 | 22.485 | 22.895 | 45.508 | 22.514 | 53.014 | 30.658 | 66.959 | 49.907 | 32.291 | 17.079 |
| MoG20_ad50_id50_mtC_r0.03 | 23.445 | 23.845 | 44.976 | 23.861 | 53.748 | 29.253 | 20.751 | 36.795 | 22.505 | 21.106 |
| MoG50_ad50_id50_mtC_r0.03 | 23.781 | 24.107 | 44.580 | 23.829 | 55.255 | 28.791 | 20.028 | 25.153 | 17.247 | 20.378 |
| MoG10_ad50_id50_mtC_r0.1 | 74.894 | 76.283 | 92.325 | 75.530 | 82.746 | 80.491 | 59.927 | 71.082 | 63.669 | 67.298 |
| MoG10_ad50_id50_mtC_r0.3 | 222.124 | 225.794 | 235.964 | 222.485 | 220.458 | 227.260 | 205.072 | 171.029 | 174.666 | 178.685 |
| MoG10_ad50_id50_mtC_r1.0 | 719.093 | 730.723 | 730.422 | 720.277 | 517.921 | 701.075 | 560.057 | 513.860 | 529.873 | 562.953 |
| MoG10_ad50_id50_mtC_r3.0 | 2099.189 | 2131.176 | 2106.525 | 2099.680 | 1518.626 | 2027.237 | 1497.029 | 1496.654 | 1483.389 | 1498.844 |
| MoG2_ad50_id50_mtR_r0.03 | 20.804 | 21.094 | 1630.390 | 20.810 | 411.953 | 149.912 | 6920.808 | 362.710 | 195.030 | 6064.530 |
| MoG5_ad50_id50_mtR_r0.03 | 21.702 | 22.051 | 3232.852 | 21.696 | 658.708 | 246.329 | 434.970 | 1011.023 | 341.312 | 430.872 |
| MoG10_ad50_id50_mtR_r0.03 | 22.487 | 22.896 | 4356.962 | 22.518 | 953.083 | 458.777 | 2980.873 | 2383.723 | 570.546 | 651.277 |
| MoG20_ad50_id50_mtR_r0.03 | 23.494 | 23.890 | 5377.364 | 1811.709 | 1113.193 | 1019.912 | 4360.021 | 2203.143 | 512.071 | 2302.820 |
| MoG50_ad50_id50_mtR_r0.03 | 25.069 | 25.495 | 6471.202 | 3583.002 | 1405.041 | 1671.400 | 4350.273 | 2992.306 | 1376.382 | 3516.809 |
| MoG10_ad50_id50_mtR_r0.1 | 74.937 | 76.321 | 4259.335 | 75.057 | 1001.369 | 496.565 | 3162.742 | 2475.862 | 471.726 | 138.372 |
| MoG10_ad50_id50_mtR_r0.3 | 224.793 | 228.979 | 4181.012 | 225.307 | 961.619 | 584.302 | 4800.571 | 1403.601 | 694.724 | 241.683 |
| MoG10_ad50_id50_mtR_r1.0 | 749.292 | 763.307 | 4351.709 | 750.725 | 1059.480 | 1100.161 | 1375.055 | 1479.128 | 827.537 | 1005.823 |
| MoG10_ad50_id50_mtR_r3.0 | 2247.890 | 2288.899 | 5581.006 | 2251.496 | 1971.971 | 2580.506 | 2141.336 | 2282.581 | 1894.595 | 1986.461 |

Table 3: Mixture distributions with 10000 training points using Hausdorff metric.

| | oracle | KDE | Gaussian | MOG | VAE | RealNVP | GAN | WGAN | WGAN-GP | GAN-NR |
|---|---|---|---|---|---|---|---|---|---|---|
| MoG2_ad50_id5_mtC_r0.03 | 2.00 | 2.00 | 2.00 | 2.00 | 2.00 | 1.99 | 2.00 | 1.99 | 2.00 | 2.00 |
| MoG5_ad50_id5_mtC_r0.03 | 5.00 | 5.00 | 4.99 | 5.00 | 4.90 | 4.93 | 4.53 | 4.92 | 4.99 | 4.99 |
| MoG10_ad50_id5_mtC_r0.03 | 9.99 | 9.98 | 9.92 | 9.98 | 9.39 | 9.76 | 9.86 | 9.91 | 9.98 | 9.98 |
| MoG20_ad50_id5_mtC_r0.03 | 19.96 | 19.91 | 18.94 | 19.75 | 12.21 | 18.84 | 19.76 | 8.13 | 19.83 | 19.91 |
| MoG50_ad50_id5_mtC_r0.03 | 49.65 | 49.75 | 41.52 | 47.61 | 18.71 | 46.89 | 49.41 | 42.50 | 49.46 | 46.40 |
| MoG10_ad50_id5_mtC_r0.1 | 9.99 | 9.98 | 9.92 | 9.98 | 9.51 | 9.69 | 9.93 | 9.43 | 9.98 | 9.92 |
| MoG10_ad50_id5_mtC_r0.3 | 9.99 | 9.98 | 9.91 | 9.98 | 9.44 | 9.81 | 9.74 | 9.73 | 9.98 | 9.99 |
| MoG10_ad50_id5_mtC_r1.0 | 9.99 | 9.98 | 9.94 | 9.99 | 9.20 | 9.78 | 9.93 | 9.79 | 9.98 | 9.99 |
| MoG10_ad50_id5_mtC_r3.0 | 9.99 | 9.98 | 9.97 | 9.99 | 9.43 | 9.73 | 9.05 | 9.95 | 9.94 | 9.95 |
| MoG2_ad50_id5_mtR_r0.03 | 2.00 | 2.00 | 2.00 | 2.00 | 2.00 | 2.00 | 1.00 | 1.10 | 2.00 | 1.00 |
| MoG5_ad50_id5_mtR_r0.03 | 5.00 | 5.00 | 4.99 | 5.00 | 5.00 | 4.96 | 4.97 | 4.81 | 4.95 | 4.97 |
| MoG10_ad50_id5_mtR_r0.03 | 9.99 | 9.98 | 9.87 | 9.99 | 9.88 | 9.84 | 8.74 | 9.97 | 9.97 | 5.73 |
| MoG20_ad50_id5_mtR_r0.03 | 19.96 | 19.91 | 19.32 | 19.94 | 19.44 | 18.80 | 9.38 | 19.41 | 19.71 | 14.91 |
| MoG50_ad50_id5_mtR_r0.03 | 49.65 | 49.75 | 41.79 | 49.44 | 47.06 | 46.01 | 17.26 | 45.32 | 48.64 | 15.49 |
| MoG10_ad50_id5_mtR_r0.1 | 9.99 | 9.98 | 9.88 | 9.99 | 9.76 | 9.84 | 8.73 | 9.67 | 9.99 | 6.86 |
| MoG10_ad50_id5_mtR_r0.3 | 9.99 | 9.98 | 9.88 | 9.99 | 9.93 | 9.87 | 8.69 | 9.88 | 9.89 | 7.58 |
| MoG10_ad50_id5_mtR_r1.0 | 9.99 | 9.98 | 9.87 | 9.98 | 9.91 | 9.30 | 7.79 | 9.92 | 9.99 | 9.19 |
| MoG10_ad50_id5_mtR_r3.0 | 9.99 | 9.98 | 9.87 | 9.97 | 9.87 | 9.51 | 7.84 | 9.93 | 9.94 | 7.46 |
| MoG2_ad50_id50_mtC_r0.03 | 2.00 | 2.00 | 2.00 | 2.00 | 2.00 | 1.99 | 1.40 | 1.87 | 1.99 | 1.98 |
| MoG5_ad50_id50_mtC_r0.03 | 5.00 | 5.00 | 5.00 | 5.00 | 4.93 | 4.97 | 4.79 | 4.97 | 5.00 | 4.99 |
| MoG10_ad50_id50_mtC_r0.03 | 9.99 | 9.98 | 9.99 | 9.98 | 8.99 | 9.87 | 9.70 | 9.87 | 8.07 | 9.92 |
| MoG20_ad50_id50_mtC_r0.03 | 19.96 | 19.90 | 19.97 | 19.91 | 18.89 | 19.69 | 19.54 | 19.94 | 19.72 | 19.87 |
| MoG50_ad50_id50_mtC_r0.03 | 49.77 | 49.68 | 49.82 | 49.56 | 47.83 | 49.47 | 48.79 | 49.64 | 46.83 | 49.77 |
| MoG10_ad50_id50_mtC_r0.1 | 9.99 | 9.98 | 9.99 | 9.97 | 9.47 | 9.94 | 9.44 | 9.81 | 9.92 | 7.09 |
| MoG10_ad50_id50_mtC_r0.3 | 9.99 | 9.98 | 9.99 | 9.98 | 5.13 | 9.88 | 9.63 | 9.31 | 9.95 | 9.65 |
| MoG10_ad50_id50_mtC_r1.0 | 9.99 | 9.99 | 9.99 | 9.98 | 9.93 | 9.98 | 9.17 | 9.97 | 9.67 | 9.37 |
| MoG10_ad50_id50_mtC_r3.0 | 9.99 | 9.99 | 10.00 | 9.99 | 9.94 | 9.98 | 9.94 | 9.28 | 9.85 | 9.95 |
| MoG2_ad50_id50_mtR_r0.03 | 2.00 | 2.00 | 2.00 | 2.00 | 2.00 | 2.00 | 1.00 | 1.99 | 2.00 | 1.00 |
| MoG5_ad50_id50_mtR_r0.03 | 5.00 | 5.00 | 5.00 | 5.00 | 4.92 | 4.97 | 4.74 | 4.99 | 5.00 | 4.77 |
| MoG10_ad50_id50_mtR_r0.03 | 9.99 | 9.98 | 9.95 | 9.98 | 9.96 | 9.94 | 7.43 | 9.78 | 9.99 | 8.33 |
| MoG20_ad50_id50_mtR_r0.03 | 19.96 | 19.91 | 19.31 | 19.93 | 19.24 | 18.75 | 8.50 | 19.87 | 19.45 | 13.30 |
| MoG50_ad50_id50_mtR_r0.03 | 49.65 | 49.75 | 40.94 | 48.41 | 46.33 | 36.63 | 26.17 | 47.89 | 44.29 | 21.10 |
| MoG10_ad50_id50_mtR_r0.1 | 9.99 | 9.98 | 9.95 | 9.99 | 9.90 | 9.87 | 7.81 | 9.87 | 9.97 | 9.37 |
| MoG10_ad50_id50_mtR_r0.3 | 9.99 | 9.98 | 9.95 | 9.98 | 9.86 | 9.86 | 4.29 | 9.96 | 9.87 | 9.53 |
| MoG10_ad50_id50_mtR_r1.0 | 9.99 | 9.98 | 9.96 | 9.99 | 9.96 | 9.78 | 9.08 | 9.85 | 9.97 | 9.76 |
| MoG10_ad50_id50_mtR_r3.0 | 9.99 | 9.98 | 9.95 | 9.98 | 9.91 | 9.76 | 9.82 | 9.92 | 9.98 | 9.95 |

Table 4: Mixture distributions with 10000 training points using mode-coverage metric.

