# OpenReview forum: "GenEval: A Benchmark Suite for Evaluating Generative Models"
_ICLR.cc/2019/Conference_

### Official Review · AnonReviewer1 · 2018-11-01
**Good but with one glaring flaw**

**Rating:** 6
**Confidence:** 4

**Review:**

Overall, this is a thorough attempt at a system for evaluating various generative models on synthetic problems vaguely representative of the kinds of problems claimed to be covered by GANs. I think the approach and the conclusions drawn are mostly reasonable, with one major caveat discussed shortly.

I also think it would help in a revision to add evaluations of more recent successors to RealNVP, such as MAF (NIPS 2017, https://arxiv.org/abs/1705.07057 ), Glow ( https://arxiv.org/abs/1807.03039 ), and (although of course this paper came out concurrently with your submission) the promising FFJORD ( https://openreview.net/forum?id=rJxgknCcK7 ). The scale of comparison of GAN variants is also much smaller than that of Lucic et al. or their followup, Kurach et al. ( https://arxiv.org/abs/1807.04720 ), which is not cited here (and should be).

But primarily, I think there are some serious concerns with your choice of metrics that make the results as they are difficult to interpret.


"Note that OT is not a distance in the standard mathematical sense, as for instance the 'distance' between two sets of points sampled from the same distribution is not zero." -- You've confused some notions here. The Wasserstein-1 distance, which is a scalar times the variant of OT you use here, absolutely is a proper distance metric between distributions: W(P, Q) is a metric. But when you compute the OT distance between *samples*, OT(S, T) with S ~ P and T ~ Q, you're equivalently computing the distance W(\hat{P}, \hat{Q}) between the empirical distributions of the samples, \hat{P} = 1/N \sum_i \delta_{S_i} and the similar \hat{Q}, which of course are not the same thing as the source distributions themselves. You can, though, view OT(S, T) as an *estimator* of W(P, Q); the distance between *distributions* is what we actually care about.

It is well-known that these empirical distributions of samples \hat{P} converge to the true distribution P (in the Wasserstein sense, W(P, \hat{P})) exponentially slowly in the dimension, which is what your example about high-dimensional distributions demonstrates. Incidentally, this is exactly the example used in Arora et al. (ICML 2017, https://arxiv.org/abs/1703.00573 ). This means that, viewed as an estimator of the true distance between distributions, the empirical-distribution OT estimator is strongly biased. Thus it becomes very difficult to tell what the true OT value is at any sample size, and moreover this amount of bias might differ for different distribution pairs even at the same sample size, so *comparing* OT estimates at a fixed sample size is a tricky business. For example, in your Figure 2, when the "oracle" score is significantly more than zero, you know that all of your estimates are very strongly biased. There is not, as far as I know, any strong reason to suspect that this amount of bias should be comparable for different distribution pairs, making any conclusions drawn from these numbers suspect.


Your scheme you call "Two-Sample Test," first, should have a more specific name. Two-sample testing is an extremely broad field, with instances including the classical Kolmogorov-Smirnov test and t tests, the popular-in-ML kernel MMD-based tests, and even Wasserstein-based tests (e.g. https://arxiv.org/abs/1509.02237 ). Previous applications of these tests in GANs and generative models include Bounliphone et al. (ICLR 2016, https://arxiv.org/abs/1511.04581 ), Lopez-Paz and Oquab (2016 - which you cite without a venue but which was at ICLR 2017), Sutherland et al. (ICLR 2017, https://arxiv.org/abs/1611.04488 ), Huang et al. (2018), and more, using a variety of schemes. Your name for this should include "nearest neighbor" or something along those lines to avoid confusion.

Also, you call this an "extension of the original formulation," but in the common case where n(x) is more often right than wrong, your v is exactly \hat t - 1 of Lopez-Paz and Oquab; see their (2). If it's usually wrong, then v = 1 - \hat t; only when the signs differ per class does it significantly differ from theirs, and in any case I don't see a real motivation to put the absolute values for each class separately rather than just taking |\hat t - 1/2|.

Moreover, it's kind of crazy to term your v statistic a two-sample *test* -- you have nothing in there about its sampling distribution, which is key to hypothesis testing to obtain e.g. a p-value. (Maybe the variance of v is very different between different distributions; this is likely the case. In any case the variance will probably become extremely large as the dimension increases.) Comparing this score is thus difficult, but in any case calling it a "test" is potentially very misleading. You could, though, estimate the variance as described by Lopez-Paz and Oquab to construct a test.

Also: you can imagine the statistic v(S, T) as an estimator of the distance between distributions given as
  D(P, Q) = |1/2 - \int ( 1 if p(x) > q(x), 0 o.w.) p(x) dx|
          + |1/2 - \int (-1 if p(x) > q(x), 0 o.w.) q(x) dx|.
But v(S, T) is, like for the OT distance, a biased estimator of this distance, whose bias will get worse with the dimension. Thus, like with the OT, it's hard to meaningfully compare v(S, T) as an attempt to compare *distributions* based on D, which is what we actually care about. Here the oracle score does not show strong bias: assuming a reasonable number of samples, when P = Q the v estimator is always going be approximately 0. But this doesn't mean that other estimators aren't strongly biased, and indeed this is exactly what your Appendix C shows. The strong change in performance for KDE is somewhat hard to interpret, but maybe has something to do with the connection between KDE and NN-based methods?


Your log-likelihood score is an unbiased and asymptotically normal estimate of the true distribution score (the cross-entropy), so it's easy to compare. But it accounts only for a very small portion of comparing distributions.


There is at least one score in common use for this kind of evaluation with easy-to-compare estimators: the squared MMD. It has an easy-to-compute unbiased and asymptotically normal estimator, so it's easy to get confidence intervals for the true value between distributions at any sample size, making comparing the numbers based on a reasonable number of samples easy. There's also a well-devolped theory for how to construct p-values for a test if you want those; Bounliphone et al. above even developed a relative test to compare the MMDs of two models accounting for the correlations due to using the same "target" set, though if you use separate target sets (because you can easily sample more points from your synthetic distribution) then it's simpler. The choice of kernel does matter, but I think the median-heuristic Gaussian kernel would be a very reasonable score to add to your repertoire, and for particular distributions you also might be able to pick a better kernel (e.g. based on the causal factors when those exist). See also Binkowski et al. (ICLR 2018, https://arxiv.org/abs/1801.01401 ) for a detailed discussion of these issues in comparison to the FID score.

Using a metric whose estimation can be understood, and whose estimators can be reliably compared, is I think vital to any evaluation process. This also prevents issues like when RealNVP outperforms the oracle, which should be impossible with any proper evaluation metric.



Minor points:

- Why is Pedregosa et al. (2011) cited for fitting multivariate Gaussians by maximum likelihood? This is something that doesn't need a citation, especially not to scikit-learn, which doesn't even (I don't think) contain an implementation of fitting Gaussians beyond (np.mean(X, axis=0), np.cov(X, rowvar=False)).

- Mode coverage and related scores: this is based on assigning sample points to their single most likely clusters? I'd imagine that sometimes a model will output points far from any cluster, in which case the cluster that happens to be closest might happen to be the most likely, but it's strange to really count that point as part of that cluster for these scores. Or similarly, a point might be relatively evenly spaced between two clusters, in which case the assignment could be fairly arbitrary, again making these scores a little strange.

---

> ### Author Response · Authors · 2018-11-26
> **Thanks for helpful review**
>
>  We thank the referee for their careful review.  Your effort has substantially improved the paper.
>
> Before we respond to specific criticisms: to the best of our knowledge, estimating the distance between continuous, high-dimensional distributions from samples in a computationally tractable manner is not a solved problem.  We actually agree with many the issues you raise with the metrics; but we feel it is better to do what we can with what we have rather than wait for the solution to this hard problem.  Already there is a lot we can say about the generative models we study even with these imperfect tools.  Moreover, we *do* put the oracle scores in each table so a reader can understand when the measurements should be regarded with caution.
>
> Responses to specific criticisms, in order of the review:
>
> "their followup, Kurach et al.": we have added a citation.  Note that this is concurrent work.
>
> "You've confused some notions here":  We have replaced the inaccurate language in the revision, we thank the reviewer for the comment.
>
> "Incidentally, this is exactly the example used in Arora et al.": we have added a citation to this example.
>
>  Note that in many of the examples where the distribution is not near a very low dimensional manifold (which is makes the estimation easier in terms of the number of samples), we use a factorized OT, projecting onto known independent sets of coordinates.  This is less powerful than the full OT, because two distributions that have the same marginals would have distance 0, but is easier to estimate.  Furthermore, we do in the text discuss the empirical limitations of OT in our particular setting, see the third paragraph in section 7.
>
> Your scheme you call "Two-Sample Test," ... ': We have changed this to "nearest-neighbor two-sample statistic.
>
> "The strong change in performance for KDE is somewhat hard to interpret, but maybe has something to do with the connection between KDE and NN-based methods?":  we added a footnote explaining this, see the bottom of page 8.
>
>
> "There is at least one score in common use for this kind of evaluation with easy-to-compare estimators... " We will add a flavor of MMD to the final revision, and run all the experiments with it as a primary distortion measure. We apologize for being unable to include it in this revision, but it will definitely be there.
>
>  A kernelized MMD with a Gaussian-type kernel is still going to be affected by the curse of dimensionality, as even though one might have better control over how the estimator converges, the metric itself becomes blurrier as a function of dimension.  However, we agree with the reviewer that the theoretical properties of MMD make it appealing and appropriate for this work, and not having MMD was a serious omission.  Again, this will be rectified for the final revision.
>
>
> "Why is Pedregosa et al. (2011) cited ":  we use this reference because we use this code
>
>
> -"Mode coverage and related scores ...": Indeed, this can fail exactly as the reviewer suggests.   Note however that a spherical Gaussian is going to have blocks of independent coordinates- it would do well on any reasonable independence test.  The point is that we consider this measurement to be not useful if the results using other metrics are very bad.

---

> > ### Comment · AnonReviewer1 · 2018-11-26
> > **Thanks for updates**
> >
> > Thanks for your response and changes in the revision; they've helped. I agree about the dimensionality issues with Gaussian-kernel MMD but also think that it will definitely help to add in a revision anyway as you've said.
> >
> > A minor note, though, that your revision has introduced several typos and grammatical errors (and also your .bib entry for Zaheer et al. is formatted incorrectly); make sure to give it a proofread in a further revision.

---

### Official Review · AnonReviewer2 · 2018-11-02
**Unsatisfactory results on a very important topic**

**Rating:** 5
**Confidence:** 4

**Review:**

This work aims at addressing the generative model evaluation problem by introducing a new benchmark evaluation suite which hosts a large array of distributions capturing different properties. The authors evaluated different generative models including VAE and various variants of the GANs on the benchmark, but the current presentation leaves the details in the dark.

The proposed benchmark and the accompanied metrics should provide additional insights about those generative models that are not well known and help drive improvement to model design, similar to [1] and [2]. But the presentation of the work, especially the experiment section, only gives abundant number of results without detailed explanation regarding the pros and cons of the existing models, the efficacy of the proposed metrics, or the reason behind some nice generative properties of GANs that are not able to learn the distribution well.

Other issues:
- In Section 1, the authors argued that "we deliberately avoid convolutional networks on images with the aim of decoupling the benefits of various modeling paradigms from domain specific neural architectures". Then in footnote 4, they mentioned that "constructed by hand neural generators that well approximate these distributions" which suggests the importance of the domain specific neural architectures. It would be nicer to see how much the "specific" neural architectures help and how different metrics favor different architectures.
- The authors only used 10K training points and 1K test samples, which seems small especially for multivariate distributions. This could have impacts on the quality of the learned models, especially the neural ones.

[1] M. Zaheer, C.-L. Li, B. Poczos, and R. Salakhutdinov. GAN connoisseur: can GANs learn simple 1D parametric distributions? NIPS Workshop on Deep Learning: Bridging Theory and Practice 2017.
[2] S. Arora, and Y. Zhang. Do GANs actually learn the distributions? An empirical study. arXiv:1706.08224.

---

> ### Author Response · Authors · 2018-11-26
> **explanation regarding the pros and cons of the existing models and the efficacy of the proposed metrics *is* substantially discussed in the paper**
>
> "are not well known":  We still see many papers suggesting that GANs learn a generator that accurately samples from a generic distribution. Just showing that this is not empirically true across a wide range of distributions and GAN variants, and with huge parameter sweeps (a grid search with about 20000 different combinations of hyper-parameters) is valuable:  even if it is well known by some,  corroboration of these results at this scale is useful.
>
> ##
>
> "Pros and Cons": we make the following conclusions, amongst others:
>
> 1: RealNVP works generally better than the other neural models we tested,  although it also fails to model distributions as simple as mixture of Gaussians with relatively large number of components and tight covariance (see the 3d paragraph in the discussion section, and also each of the paragraphs in section 7 describing the tables).
>
> 2: Mixture of Gaussians and KDE are baselines that can be hard to beat in these simple settings,  but can also fail catastrophically, e.g. for MOG when the number of clusters in the data is much higher than in the mixture (figure 1 and 2), and for KDE when distributions have product structure (figure 4 and the discussion of it in section 7)
>
> 3: different distortion statistics offer complementary insights into the performance of various models. In particular, while 2S is generally more trustworthy since its oracle performance is closer to 0, OT can offer a different perspective (see 3d paragraph in section 7 describing figure 1). This is true even when OT "fails" (see VAE working better than oracle hinting at its denoising effect).   Specifically, OT is less trustworthy when the true distribution has an intrinsically high dimension, and fills space.
>
> 4: The GAN variants do not generally seem to improve over vanilla GAN, except in cluster coverage in the product of mixtures distribution (discussed in the 2nd paragraph of the discussion section).
>
> 5: VAE's can collapse to a manifold, and fail at capturing "noise" dimensions (see 4th paragraph in section 7 or last paragraph in the discussion section 8)
>
> Our paper documents an analysis tool designed to answer the question "Can your model learn to sample some simple distributions?".   We absolutely think it will help drive model design, because if you think, for example, that your new optimization method is making GANs better, you can test that.   On the other hand,  we think explaining the nice properties of GANs, or elucidating "the reason ...  nice generative properties of GANs ..."  is beyond the scope of this work.    As is making specific recommendations on how to improve GANs is outside the scope of the work.  These are interesting topics, but they are not the topics of this paper.
>
> ##
>
> w.r.t. [1] and [2]:  our work can be considered a far more comprehensive version of [1]; and in particular, [1] does not really offers any insights that help drive model design, except to show that some forms of GAN seem to struggle with simple 1d distributions.  [2] is demonstrating the results of a *particular* test to see if a generative model learns the input distribution, applies it to the setting of image-generating models, and concludes that GANs that produce high quality images are not learning the distribution.  That work does suggest a method for improving model design for GAN, namely increasing the discriminator capacity; but in our work we see that this does not seem to be enough.  Again, our work is a much more comprehensive suite of tests, and the results of those tests on a set of popular generative modeling protocols.  We thank the reviewer for reminding us of these; we cite them in the related work.

---

> ### Author Response · Authors · 2018-11-26
> **response to "other issues"**
>
> ##
>
> " - In Section 1, the authors argued that "we deliberately  ...  domain specific neural architectures". Then in footnote 4, they mentioned that ....":
>
> The "specific neural architectures" that can generate the distributions *were* in the search space of the parameter sweeps/optimizations; that is the point of the comment.   We explicitly wrote down the weights of a network (with a choice of hyper-parameters from the sweep) that was a (near) solution to the given modeling problem.  The goal of this exercise was to show that the modeling failures were not because it is impossible to model the distributions with the architectures that were in the sweep, but because the protocol/optimization failed to find the correct parameters.
>
> ##
>
> "The authors only used ..." :
>
> The distributions should be learnable with 10K training points.  They either have very low-dim structure (2 or 3 intrinsic dim), a small number of very well defined clusters (<50),   or independence structure that factorizes the distribution into several (simpler) distributions with these structures (e.g. the product of mixtures is 3 clusters per independent factor, with <8 factors). The empirical results suggest that these *are* mostly learnable at this numbers of samples, even if not learned by every scheme.   Furthermore, sample complexity is a reasonable thing to care about- in our opinion restricting to the case where one has access to unlimited samples is unrealistic.

---

> > ### Comment · AnonReviewer2 · 2018-11-27
> > **Further questions**
> >
> > I am a bit confused here. Why did the protocol/optimization fail to find the correct parameters from the sweep if the best choice of hyper-parameters is in the sweep space? I assume the authors were referring to optimize towards different metrics.

---

> > > ### Author Response · Authors · 2018-11-28
> > > **failure of optimization/training protocol**
> > >
> > > We mean that for a distribution P, we can write down the neural generator (i.e. explicitly write down the values of the weights) that would map Gaussian noise to P with low distortion in our metrics.  However, when we train a GAN, for example, with the generator hyperparameters (e.g. number of layers, nonlinearity type, and hidden dimensions) of the hand-designed model (and a sweep over discriminator parameters),  the training fails to find a generator that is low distortion.  Thus we know that the reason for the failure to find the generator is because of the optimization/optimization protocol, and not because a good generator is not realizable by the neural nets we use.

---

### Official Review · AnonReviewer3 · 2018-11-06
**This paper proposes a series of metrics and generative models to evaluate different approximate inference frameworks.**

**Rating:** 5
**Confidence:** 3

**Review:**

Updated to reflect author response:

This paper proposes a series of metrics to use with  a collection of generative models to evaluate different approximate inference frameworks. The generative models are designed to be synthetic and not specialized to a particular task. The paper is clearly written and the motivation is very clear.

While there has been work like Forestdb to maintain a collection of generative models, I don't believe
there has been work to evaluate how they perform on a series of metrics. There would be great utility
in having a less ad-hoc way to evaluate inference algorithms.

While the idea is sound, the work still feels a bit incomplete. The only distributions used in the experimental section seem to be Gaussians and Mixture of Gaussians. Many more families of distributions are mentioned in Section 3, and it would have been nice to show some evaluation of them considering the code is already there. In addition to distributions mentioned in Section 3, it would help if there were a few larger dimensional distributions. Often for evaluation now, many papers
use a Deep Gaussian model trained to model MNIST digits. I worry that insights drawn from
the synthetic examples won't transfer when the models are applied to real-world tasks.

I would like to see described a wider variety of models, including possibly more models with
discrete latent variables as much recent literature is currently exploring.

The paper is a bit confusing in how it discusses distributions and models. Distributions form the ground truth we compare different trained models to.  It would been more clear for me if the explanation with supplemented with some notation to describe who will compare draws from the true data distributions to samples from each of the trained generative models.

---

> ### Author Response · Authors · 2018-11-26
> **rebuttal:**
>
> “The only models mentioned are Gaussians and Mixture of Gaussians. Section 4 mentions VAEs and GANs but those are not generally seen as particular generative models.” :
> This is not true.  We show results using Gaussians, Mixtures of Gaussians, Kernel Density Estimators, VAEs, several flavors of GANs, and Real NVPs (see sec. 4).
> While one can make an argument that a GAN is not a generative model by some definitions, all the other models are generative models by standard definitions.  Furthermore, we take pains to give a definition of “generative model” as used in the paper (see third paragraph in the introduction).
>
>  “I would like to see described a wider variety of models, including possibly models with discrete latent variables as much recent literatue is currently exploring.”:
> Besides the fact that we do in fact have a wide variety of deep and non-deep models, note that we include mixtures of Gaussians, and KDE’s, both of which have discrete latent variables.
>
> ““Often for evaluation now, many papers use a Deep Gaussian model trained to model MNIST digits. I worry that insights drawn from the synthetic examples won't transfer when the models are applied to real-world tasks.”
> The point of this work is to work in a controlled setting. Since we know the properties of the ground truth data distribution, we can leverage these properties to much more accurately estimate model fitting using our various distortion metrics. See last paragraph of sec 7 and sec. 8 for a summary of the findings and contributions. In short, this work is meant to be complementary to prior attempts at comparing generative models using natural images.

---

> > ### Comment · AnonReviewer3 · 2018-12-04
> > **Response to rebuttal**
> >
> > Thank you for your thoughtful reply. I have updated my review to take into account your response. One clarification is that I think you have a wonderful variety of generative models, but I wish there were more distributions. Many of these models only sense on more complex distributions and it's worth creating some larger synthetic ones to capture some of what's going on larger settings.

---

> > > ### Author Response · Authors · 2018-12-04
> > > **thanks for the response**
> > >
> > >  w.r.t.  "Many of these models only sense on more complex distributions": Choosing synthetic distributions with easily understood properties is part of the contribution of this work.  Our approach allows analysis of which models can approximate distributions with which kinds of properties.
> > >
> > >  While we agree with the statement that on simpler distributions, the neural models may be overkill, we think the idea that a model should *only* "work" when the problem is hard (and hard to measure success on) to be a cause for concern.  We are careful in this work to not suggest that simple models that can do well on these distributions are better models (see our discussion section).  On the other hand, we do see that the neural models often fail to successfully approximate simple distributions.   One may surmise that even though these models can generate interesting and convincing samples in more complex settings (e.g. with convolutional networks and image data), they are probably not approximating the distribution there either (corroborating several other works suggesting this).
> > >
> > >  Note that if nothing else, decoupling the influence of the inductive biases of the model architecture and the modeling protocol is important.

---

> > > > ### Comment · AnonReviewer3 · 2018-12-08
> > > > **The value of simple examples**
> > > >
> > > > I completely agree on the value of simple distributions. Something is fundamentally wrong or at least very illustrative if a method demonstrates poor performance on a simple distribution. It's crucial for simple distributions to be included. I think the value of a benchmark suite is to strike this balance between being focused and being somewhat comprehensive. It also helps if it can be connected to previous results on more complex distributions. I think for a benchmark suite to be really valuable it has to some examples of both.

---

> > > > > ### Author Response · Authors · 2018-12-08
> > > > > **more complex distributions**
> > > > >
> > > > > Note that we do have "image-like" distributions in the set (the "shifted bumps" distribution).  Moreover, all the distributions we show results for have parameterized difficulty:  for example, with the shifted bumps, it is the size of the image (equivalently, the number of bumps), the random scalings to the height and width of the bump, and the amount of ambient noise.    It is  trivial to adjust the parameters to make the problem harder if we want.
> > > > >
> > > > > When we use a mixture of Gaussians, the difficulty depends on the number of components, and the covariance structure of the components.  We show in Figure 1 the effect of changing these.
> > > > >
> > > > > When we build a product of mixtures of Gaussians, the complexity depends on the number of mixture components c per product component, and the number p of product components.  We show two settings for these in Figure 6.
> > > > >
> > > > > Finally, note that it is also trivial with the toolbox to compose distributions to get something as complicated as you want.  You want a product of mixtures of shifted bumps and low-d manifolds?  its easy, just a line of code.   The requirement we have is that any distribution we add we have full control over: we should be able to compute its log-likelihood, sample from it, etc.; but by composing the building blocks one can make enormously complicated distributions over which we have full control.  We don't display the results of these things because we don't consider them informative per amount of space we are allowed.
> > > > >
> > > > > So, in short:  the tool makes it easy to control the difficulty of the problem.  The tool does have image-like distributions.   We show results on complexities that we consider informative, but this is not because a limit of the tool, but rather because a limit on the pages in a submission.

---

### Meta-Review · Area_Chair1 · 2018-12-14
**Intersting benchmark suite that could be extended.**

**Confidence:** 3
**Recommendation:** Reject

**Metareview:**

The paper introduces a benchmark suite providing a series of synthetic distributions and metrics for the evaluation of generative models. While providing such a tool-kit is interesting and helpful and it extends existing approaches for evaluating generative models on simple distributions, it seems not to allow for very different additional conclusions or insights.This limits the paper's significance. Adding more problems and metrics to the benchmark suite would make it more convincing.